# Tuberculosis-related deaths at a tertiary hospital in Zambia: Insights into the prevalence and associated factors

**Lukundo Siame**, **Eemmanuel Chembe**, **Lweendo Muchaili**, **Benson M. Hamooya**, **Sepiso K. Masenga**\*

School of Medicine and Healt3h Sciences, Mulun3gushi University, Livingstone, Zambia

\* sepisomasenga@gmail.com

**Data Availability Statement:** The data underlying the results presented in the study have been provided as supporting information.

## Abstract

Tuberculosis (TB) mortality remains a significant public health concern globally. This study aimed to determine the prevalence of tuberculosis-related deaths and associated factors among patients at Livingstone University Teaching Hospital (LUTH) Chest Clinic, Zambia. We conducted a retrospective cross-sectional study among 694 individuals (507 adult and 187 children) diagnosed with drug susceptible TB disease between January 1, 2021, and December 31, 2022. Demographic and clinical information were collected from medical records using a data collection form. Multivariable logistic regression was used to determine factors associated with TB-related death. Statistical significance was set at p < 0.05. STATA version 15 was used for all data analysis. The prevalence of TB-related death among adults (above 19 years old) was 18.4% (n = 93) whereas that in children (below 19 years old) was 7.0% (n = 187). Living with HIV (AOR 1.75, 95% CI 1.00–3.08, p = 0.049) was positively associated with TB-related death among the adult patients while being on a family based direct observation therapy (DOT) plan was negatively associated with TB-related death both among adults and among children, (AOR 0.24, 95% CI 0.13–0.45, p <0.001) and (AOR 0.2, 95% CI 0.03–0.99, p = 0.039) respectively. This study found a high TB-related mortality rate, both among adults and children, exceeding the national target of 5% and it was significantly associated with HIV status and DOT plan. There is therefore a need to enhance strategies aimed at reducing TB-related deaths, especially among those living with HIV.

## Introduction

Tuberculosis (TB) remains a significant global health challenge, with an estimated 1.3 million deaths among HIV-negative individuals and 214,000 deaths among those co-infected with HIV in 2021 [1]. The World Health Organization (WHO) has established goals to drastically reduce TB deaths and cases by 2035 [2, 3]. However, the COVID-19 pandemic caused a major setback, with a 4.5% increase in TB deaths in 2020 due to disruptions in TB services and decreased access to diagnosis and treatment [4, 5].

**Funding:** The authors received no specific funding for this work.

**Competing interests:** The authors have declared that no competing interests exist.

Zambia, a high TB-burden country, faces a significant challenge. In 2020, it was estimated to have 59,000 TB cases, ranking 21st among the 30 countries with the highest TB burden [6]. While there has been progress, with the overall TB mortality rate (excluding HIV co-infection) decreasing from 30 to 21 per 100,000 population between 2015 and 2021, a lot of work remains [6]. This decline is mirrored in the decreasing mortality rate among persons living with HIV, from 84 to 21 per 100,000 in the same period [6].

Several established risk factors contribute to TB mortality globally, including malnutrition, diabetes mellitus, previous TB treatment, anemia, and substance abuse [7, 8]. Studies have identified additional factors such as liver cirrhosis, bacterial pneumonia, advanced age, repeated anti-TB treatment, relying solely on clinical diagnosis (without confirmatory tests), and unknown HIV status [7, 8]. Notably, extra-pulmonary TB (EPTB) is associated with a higher risk of death compared to pulmonary TB (PTB) [7, 8]. In resource-limited settings like Zambia, most TB deaths occur within the first two months of treatment, often linked to delayed diagnosis or unidentified comorbidities [7, 8].

To effectively combat TB mortality in Zambia, a deeper understanding of the factors associated with TB in our specific context is crucial. This study aimed to determine the prevalence of tuberculosis-related death among adults and children and identify the factors associated with death among TB patients at Livingstone University Teaching Hospital Chest clinic, Zambia.

## Methods

### Study design and setting

This was a retrospective cross-sectional study which utilized programmatic data from the Chest Clinic between January 1, 2021, and December 31, 2022 at Livingstone University Teaching Hospital (LUTH), with the largest Tuberculosis Treatment center in the Southern Province of Zambia.

The clinic is the primary department, which manages all patients of all ages diagnosed with TB and leprosy. A physician leads the clinic, with assistance from two medical assistants and two nurses. The unit receives patients with TB diagnoses from various departments within the hospital, including internal medicine, pediatrics, surgery, obstetrics and gynecology, as well as patients from other facilities. The presumed TB register records individuals suspected of TB, collects their specimens (sputum, stool, urine, pleural fluid, ascites fluid, and others), and sends the samples to the laboratory for examination. The clinic receives feedback from these tests. Those diagnosed with TB are then entered in the TB treatment register, as either drug-susceptible TB or drug-resistant TB, and treated based on recommended guidelines [9]. Outcomes are recorded in the TB treatment and patients' cards.

### Study participant and sampling methods

Participants were excluded if the treatment outcome could not be determined, and the age was missing from the treatment register.

### Variables

The main outcome was death classified as binary (died or didn't die) and the independent variables which were considered were demographics (age, sex, residence, occupation), HIV status, type of TB, treatment regime, presumptive co-trimoxazole therapy given to people living with HIV, type of Patient (New, relapse, Treatment after lost to follow-up, other Category), blood test (platelet count, hemoglobin, white blood cell count, CD4 count, liver enzymes (alanine transaminase, aspartate transaminase).

## Data collection

Data was collected by trained research assistants who abstracted data from paper-based drug susceptible TB treatment register and then completeness of information of the records was audited for data accuracy at the end of each data collection session to ensure completeness of the data. The data was collected between January 6[th], 2023, and January 24[th], 2023.

## Operation definition

This study adopted World Health Organization (WHO) guidelines to define TB cases, treatment outcomes, and patient types [10].

1. **TB Cases:** A confirmed TB case required either a positive biological specimen (smear microscopy, culture, or GeneXpert) or a clinical diagnosis by a healthcare professional who initiated a full course of TB treatment [11].

2. **Treatment Outcomes:** Successful outcomes were defined as the sum of "cured" and "treatment completed". A "cured" case involved a patient with bacteriologically confirmed pulmonary TB at the beginning of treatment who completed treatment as recommended by national policy with evidence of bacteriological response and no evidence of failure [10]. "Treatment completed" indicated a patient who completed treatment as recommended by the national policy whose outcome does not meet the definition for cure or treatment failure [10]. Unsuccessful outcomes included treatment failure, default, or death. "Died" referred to any TB patient who died before initiating or during the treatment course. "Lost to follow-up" indicated a patient who either never started treatment or had a treatment interruption for two consecutive months or more [10]. Treatment failure involved a patient whose treatment regimen needed to be terminated or permanently changed to a new regimen or treatment strategy [10].

3. **Patient Types:** New patients had never received TB treatment or taken anti-tuberculosis drugs for less than a month. Previously treated patients had received anti-TB drugs for at least one month in the past and were further categorized based on their most recent treatment outcome [11]:

   a. Relapse: Patients previously treated for TB, declared cured or treatment completed, who now had a recurrent episode (relapse or reinfection).

   b. Treatment after Loss to Follow-up: Patients previously treated for TB who were declared lost to follow-up at the end of their most recent treatment course. (Previously known as "treatment after default").

   c. Other category: Patients who did not fit into any of the above categories.

## Data analysis

Data was first abstracted from paper-based TB registers then entered on paper-based data collection tools. Data were entered into the REDCap application, cleaned in Excel, and then exported to STATA version 15 for data analysis. To test for normality Shapiro Wilk's test and histograms were used. Data was described using frequency and percentages for categorical variables and median (interquartile range) was used for continuous variables. The Wilcoxon rank-sum test was used to compare the statistical difference between two medians. A relationship between two categorical variables was determined using a chi-squared test. The Hosmer-Lemeshow test was used to determine how well the model fits before logistic analysis.

Multivariable and univariable logistic regression was used to examine the factors associated with tuberculosis death using Generalized Linear Models equations (GLM) via the logit link function to account for correlation structure within observations. Statistical significance was set at $p < 0.05$.

## Ethics

Ethical approval to conduct this study was obtained on 19[th] May 2022 from Mulungushi University Research Ethics Committee (SMHS-MU2-2022-48) for a period 19[th] May 2022 to 19[th] May 2023. All data analyzed were de-identified to ensure confidentiality such that no patient was identifiable during or after the data collection. Due to the use of secondary data from existing patient registers and files, the need for informed participant consent was waived by the relevant ethics committee.

We have used the Strengthening the Reporting of Observational Studies in Epidemiology (STROBE) guidelines for reporting this observational study (S1 Checklist).

## Results

Out of the 719 patient records, only 694 were eligible for inclusion of which 507 were adults and 187 were children, **Fig 1**.

### Basic characteristics

Of 507 adult participants, the median age was 41 (interquartile range (IQR): 32, 51) with the age group 35–44 years mostly affected (n = 147, 29%), **Table 1**. The majority were males (n = 276, 54%). The majority of the participants were from urban areas (n = 293, 57.8%). Few participants were miner (n = 8, 1.6%) and healthcare workers (n = 6, 1.2%). People living with HIV were more than those without (n = 311, 61.5%). The majority of individuals had been diagnosed with pulmonary tuberculosis (TB) (n = 308, 69.8%), and all participants were on first-line treatment. The majority of the participants had received Cotrimoxazole preventive therapy (CPT) (n = 290, 93.3%). Most TB cases, 73.1% (n = 370), were newly diagnosed cases. The median weight at the start and end of treatment was 48 kg (IQR: 43, 59) and 58 kg (IQR: 50, 67), respectively. The direct observation plan for most participants was at the hospital (n = 270, 56.0%). 18.3% (n = 93) of the study participants died. The median Hemoglobin (HB) level was 10.9 g/dL (IQR: 8.6, 12.8), while the median white blood cell count was 5.9 x $10^9/L$ (IQR: 4.2, 8.6). The median creatinine and urea for the participants were 92.1 $\mu mol/L$ (IQR: 80.2, 119.2) and 3.8 mmol/l (IQR: 2.9, 5.4), respectively. The median CD4 count, HIV Viral load, white blood cells, Alanine transaminase, Aspartate transaminase, and Platelets were 374 *cells/μL* (IQR:162, 566), 30 copies/ml (IQR:0, 1310), 5.9 x $10^9/L$ (IQR: 4.2, 8.6), 22.4 IU/L (IQR: 15, 33.2), 34.6 IU/L (IQR: 27.8, 48.8), and 266 x $10^9/L$ (IQR: 189, 382), respectively.

Of the 187 pediatrics patients, the median age was 2 years (IQR: 1, 8) with the under 5 age group mostly affected (n = 115, 61.5%) and 55.6% (n = 104) were male. The percentage of children coming from urban areas (n = 94, 50.3%) and rural areas (n = 93, 49.7%) was nearly the same. Only about 22.9% were living with HIV. Majority of the participant had received CPT (n = 36, 90.0%) and were newly diagnosed TB cases (n = 169, 90.4%) **Table 2**.

### Relationship between TB-related death and socio and demographic characteristics of participants

The overall prevalence of TB-related death among adults was 18.3% (n = 93), **Table 1**. However, the prevalence was lower when compared to those with pulmonary TB (n = 43/ 507,

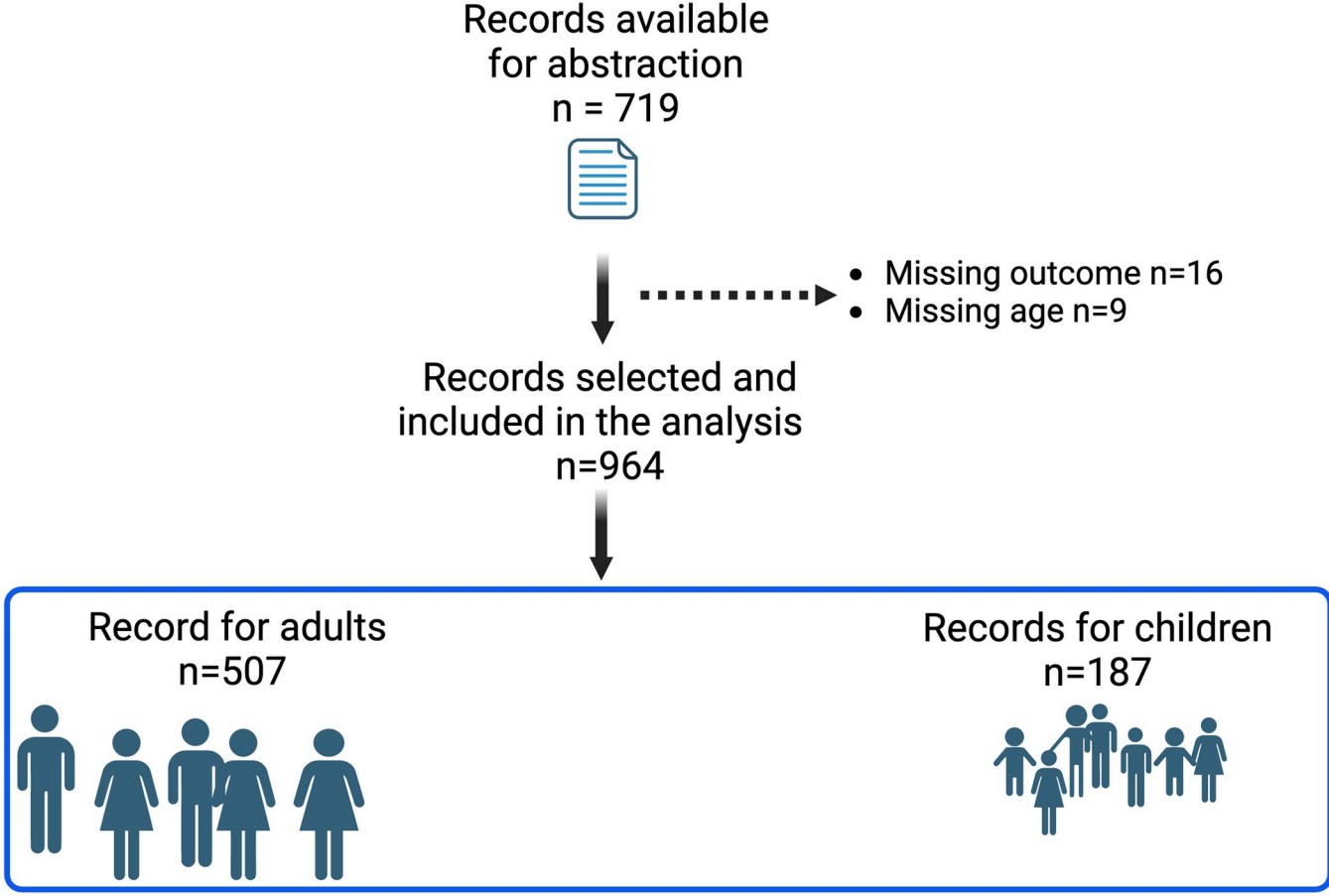

**Fig 1. Screening of eligible participants.** Abstracted records were 719 in total. We excluded 16 files with missing outcome data and 9 files with missing age and included 964 files in the final analysis (507 adults and 187 children).

8.3%) and extra pulmonary TB (n = 51/507, 10.1%), **S1 Table**. A majority of the deaths occurred in people living with HIV as compared to those who were HIV-negative (21.5% vs. 13.3%). More people died under clinic-observed patients as compared to those who were on family based direct observation therapy (DOT) plan (25.2% vs. 8.9%). The majority of participants who died had a low HB (9.2 vs. 11.3 g/dl) and CD4 cell count (160.5 vs. 386 *cells/μL)* compared to those who survived. Individuals who died had a higher urea level than those who didn't (6.5 vs. 3.8 mmol/l). A higher proportion of patients who were observed under the clinic-based plan died when compared to those under the family-based plan for those with pulmonary TB only (11.5% vs. 4.3%) and those with extra pulmonary TB (13.7% vs. 4.7%) respectively, **Table 2**.

The overall prevalence of TB-related death among pediatric patients was 7.0% (n = 13), **Table 2**. On the other hand, the proportion of TB-related death among those who had only pulmonary TB and extra pulmonary TB was 3.7%(n = 7) and 6.2% (n = 6) respectively, **S2 Table**. The majority of children who died were in the age group 12–18 years old compared to other age group (17.1% vs. 5.4% vs. 4.4%). The majority of participants who died were living with HIV when compared to those without HIV (17.5% vs. 4.4%). Among those who were on cotrimoxazole therapy, the majority didn't die when compared to those who died (88.9% vs. 11.1%), **Table 2**.

**Table 1. Basic demographic and clinical characteristics of adult patients with drug-susceptible TB mortality (N = 507).**

| Variable | N | Median, (IQR) OR Frequency (%) | Died (n = 93, 18.4%) | Alive (n = 414, 81.6%) | P value |
|---|---|---|---|---|---|
| **Age,** years | 507 | 41 (32, 51) | 41 (34, 51) | 41 (32, 51) | 0.559 |
| **Age category,** years | 507 | | | | 0.579 |
| 19–24 | | 50 (9.9) | 5 (10.0) | 45 (90.0) | |
| 25–34 | | 112 (22.1) | 20 (17.9) | 92 (82.1) | |
| 35–44 | | 147 (29.0) | 30 (20.4) | 117 (79.6) | |
| 45–54 | | 101 (20.0) | 22 (21.8) | 79 (78.2) | |
| 55–64 | | 44 (8.0) | 7 (15.9) | 37 (84.1) | |
| 65 and above | | 53 (10.0) | 9 (17.0) | 44 (83.0) | |
| **Sex** | 507 | | | | |
| Male | | 276 (54.4) | 53 (19.2) | 223 (80.8) | |
| Female | | 231 (45.6) | 40 (17.3) | 191 (82.7) | |
| **Residence** | 507 | | | | 0.771 |
| Urban | | 293 (57.8) | 55 (18.8) | 238 (81.2) | |
| Rural | | 214 (42.2) | 38 (17.8) | 176 (82.2) | |
| **Occupation** | 507 | | | | 1 |
| Health care worker | | 6 (1.2) | 1 (16.7) | 5 (83.3) | |
| Miner | | 8 (1.6) | 1 (12.5) | 7 (87.5) | |
| Others | | 493 (97.2) | 91 (18.5) | 402 (81.5) | |
| **PLWH,** | 506 | | | | **0.02** |
| Yes | | 195 (38.5) | 67 (21.5) | 244 (78.5) | |
| No | | 311 (61.5) | 26 (13.3) | 169 (86.7) | |
| **Presumptive CPT,** | 311 | | | | 0.624 |
| Yes | | 290 (93.3) | 61 (21.0) | 229 (79.0) | |
| No | | 16 (5.1) | 5 (31.3) | 11 (68.7) | |
| missing/unknown | | 5 (1.6) | 1 (20.0) | 4 (80.0) | |
| **Patient type** | 506 | | | | 0.489 |
| New | | 370 (73.1) | 67 (18.1) | 303 (81.9) | |
| Relapse | | 99 (19.6) | 22 (22.2) | 77 (77.8) | |
| Treatment after loss to follow up | | 20 (4.0) | 2 (10.0) | 18 (90.0) | |
| other category | | 17 (3.4) | 2 (11.8) | 15 (88.2) | |
| **Type of TB** | 507 | | | | |
| Pulmonary | | 308 (69.8) | | | |
| Extra pulmonary TB | | 199 (39.2) | | | |
| **Weight at Start of treatment, kg,** | 491 | 53.7 (45, 61) | 48 (43, 59) | 54 (46, 62) | **0.002** |
| **Weight at end of treatment,kg,** | 361 | 59.6 (50, 67) | 45.5 (40, 62) | 58 (50, 67) | 0.119 |
| **DOT plan,** | 482 | | | | **< 0.001** |
| Observed daily at clinic | | 270 (56.0) | 68 (25.2) | 202 (74.8) | |
| Observed daily by family | | 212 (44.0) | 19 (8.9) | 193 (91.0) | |
| **Treatment outcome** | 507 | | | | |
| Cured | | 409 (80.7) | | | |
| Died | | 93 (18.3) | | | |
| lost to follow up | | 5 (1.0) | | | |
| Treatment failure | | 0 (0.0) | | | |
| **HB, *g/dL*,** | 104 | 10.9 (8.6, 12.8) | 9.2 (7.2, 10.4) | 11.3 (9.2, 13.1) | **< 0.001** |
| **WBC, *10^9/L*,** | 177 | 5.9 (4.2, 8.6) | 7.35 (4.5, 10.0) | 5.8 (4.1, 8.1) | 0.097 |
| **Creatinine, *μmol/L*,** | 99 | 92.1 (80.2, 119.2) | 110.7 (72, 161.6) | 91.8 (81.4, 110) | 0.408 |

(*Continued*)

**Table 1.** (Continued)

| Variable | N | Median, (IQR) OR Frequency (%) | Died | Alive | P value |
|---|---|---|---|---|---|
| | | | (n = 93, 18.4%) | (n = 414, 81.6%) | |
| **Urea,** *mmol/L,* | 73 | 4.1 (3.1, 6.5) | 6.5 (4.9, 12.2) | 3.8 (2.9, 5.4) | **0.002** |
| **Platelet,** *10^9/L,* | 174 | 266 (189, 382) | 264.9 (185, 322) | 270.5 (193, 396) | 0.241 |
| **ALT,** *IU/L,* | 63 | 22.4 (15, 33.2) | 24.8 (14.2, 30.2) | 21.4 (15.2, 33.2) | 0.945 |
| **AST,** *IU/L,* | 80 | 34.6 (27.8, 48.8) | 44.4 (31.9, 54.1) | 32.4 (26.4, 46.5) | 0.064 |
| **CD4 count,** *cells/μL,* | 71 | 374 (162, 566) | 160.5 (37, 400) | 386 (22.3, 604) | **0.008** |
| **HIV VL, copies,** | 20 | 30 (0, 1310) | 15 (0, 1025) | 30 (0, 1775.5) | 0.467 |

Note: 204 TB cases were diagnosed clinically and the rest were bacteriologically confirmed

Abbreviation: HB (hemoglobin),CPT (Cotrimoxazole preventive therapy), WBC (white blood cells), ALT (Alanine transaminase), AST(Aspartate transaminase), DOT (Direct observation therapy, others (other profession category), TB (tuberculosis), PLWH (People living with HIV), HIV VL (HIV viral load)

### Multivariable and univariable analysis of factors associated with TB related deaths

Table 3 shows factors associated with TB related deaths on univariable and multivariable logistic regression among adult patients. On univariable analysis, people living with HIV were 1.78 times more likely to die than those without HIV. Participants whose medications were observed by family based direct DOT plan had a 71% likelihood of not dying compared to those observed by facility-based DOT plan. On multivariable analysis, people living with HIV were 1.75 times more likely to die than those without HIV, while being on family based direct observation therapy (DOT) plan was associated 76% reduced odds of dying with drug-susceptible TB. The odds of dying for pulmonary TB and Disseminated TB were similarly reduced for those on famil-based DOT, S3 Table.

Table 4 shows factors associated with TB related deaths at univariable and multivariable logistic regression among pediatric patients. At univariable analysis, the children in the age group 12–18 years old had 4.5 increased odds of dying when compared to the age group 0–4 years old. People living with HIV were 4.6 times more likely to die than those without. At multivariable analysis, the children in the age group 12–18 years old had 4.9 times odds of dying when compared to the age group 0–4 years old. While being on family based direct observation therapy (DOT) plan was associated 60% reduced odds of dying. When segregated by type of TB, only the age category 12–18 years, living with HIV and use of family-based DOT were associated with death in children with pulmonary TB while disseminated TB abrogated this relationship, S4 Table.

### Discussion

This study investigated the prevalence of tuberculosis-related death and associated factors among pediatrics and adult patients diagnosed with TB at the Livingstone University Teaching Hospital chest clinic. Our study found a prevalence of tuberculosis-related death of 18.3% among adults in our facility, this is higher than a study conducted in Zambia (2021) which found a prevalence of 10.3% [12]. On the other hand, the prevalence of tuberculosis-related death among pediatric patients was 7.0%, which was lower than the global prevalence of 14% among HIV-negative children and 11% among children living with HIV in 2021 [13]. The burden remains high in our setting and this can be attributed to LUTH being a referral hospital that attends to a high volume of patients, shortage of trained personnel, scarce diagnostic resources like GeneXpert leading to delays in TB diagnosis, lack of robust follow up

**Table 2. Basic demographic and clinical characteristics of pediatric patients with drug-susceptible TB mortality (N = 187).**

| Variable | N | Median, (IQR) OR Frequency (%) | Died | Alive | P value |
|---|---|---|---|---|---|
| | | | 13 (7.0%) | 174 (93.0) | |
| **Age,** *years* | 187 | 2 (1, 8) | 10 (1, 16) | 2 (1, 7) | 0.106 |
| **Age category,** *years* | 187 | | | | **0.031** |
| 0–4 | | 115 (61.5) | 5 (4.4) | 110 (95.7) | |
| 5–11 | | 37 (19.8) | 2 (5.4) | 35 (94.6) | |
| 12–18 | | 35 (18.7) | 6 (17.1) | 29 (82.9) | |
| **Sex** | 187 | | | | 0.894 |
| Male | | 104 (55.6) | 7 (6.7) | 97 (93.3) | |
| Female | | 83 (44.4) | 6 (7.2) | 77 (92.7) | |
| **Residence** | 187 | | | | 0.377 |
| Urban | | 94 (50.3) | 5 (5.3) | 89 (94.7) | |
| Rural | | 93 (49.7) | 8 (8.6) | 85 (91.4) | |
| **PLWH** | 175 | | | | **0.006** |
| Yes | | 40 (22.9) | 7 (17.5) | 33 (82.5) | |
| No | | 135 (77.1) | 6 (4.4) | 129 (95.6) | |
| **Presumptive CPT** | 40 | | | | **< 0.001** |
| Yes | | 36 (90.0) | 4 (11.1) | 32 (88.9) | |
| No | | 1 (2.5) | 0 (0.0) | 1 (100.0) | |
| missing/unknown | | 3 (7.5) | 3 (100.0) | 0 (0.0) | |
| **Patient type** | 187 | | | | 0.895 |
| New | | 169 (90.4) | 12 (7.1) | 157 (92.9) | |
| Relapse | | 11 (5.9) | 1 (9.1) | 10 (90.9 | |
| Treatment after loss to follow up | | 3 (1.6) | 0 (0.0) | 3 (100.0) | |
| other category | | 4 (2.1) | 0 (0.0) | 4 (100.0) | |
| **Type of TB** | 187 | | | | 0.315 |
| Pulmonary | | 76 (40.6) | 7 (9.2) | 69 (90.7) | |
| Extra pulmonary | | 111(59.4) | 6 (5.4) | 105 (94.6) | |
| **Weight at Start of treatment, kg,** | 185 | 9.2 (6.8, 19.8) | 25 (7.0, 38.5) | 9.2 (6.8, 19) | 0.245 |
| **DOT plan** | 184 | | | | 0.207 |
| Observed daily at clinic | | 127 (69.0) | 11 (8.7) | 116 (91.3) | |
| Observed daily by family | | 57 (31.0) | 2 (3.5) | 55 (96.5) | |
| **Treatment outcome** | 187 | | | | |
| Cured | | 173 (92.5) | | | |
| Died | | 13 (7.0) | | | |
| lost to follow up | | 0 (0) | | | |
| Treatment failure | | 1 (0.5) | | | |

**Abbreviation:** DOT (Direct observation therapy, TB (tuberculosis), PLWH (People living with HIV), CPT (cotrimoxazole)

**Note**: 203 TB cases were diagnosed clinically and the rest were bacteriologically confirmed, cotrimoxazole was given to HIV patient only

mechanism and defaulting on antiretroviral therapy medication by persons living with HIV who mostly present with a severe form of TB leading to poor prognosis.

This present study found that compared to cases aged 0–4 years the odds of dying from TB was higher among the cases aged 12–18 years. This finding is similar to two studies conducted in south Africa in 2019 and 2021 [14, 15]. On the other hand, most studies have shown that children under age of 5 years are mostly affected contrary to our study [16]. The probable reason for this discrepancy could be due to late presentation and TB/HIV co-infection which might be the contributing factor in our setting [17].

**Table 3. Univariable and multivariable logistic regression of factors associated with drug-susceptible TB mortality among adult patients.**

| Variable | OR (95%) | P-value | AOR (95%, Cl) | P-value |
|---|---|---|---|---|
| **Age, years** | 1.00 (0.99, 1.02) | 0.748 | 1.00(0.98, 1.02) | 0.787 |
| **Sex** | | | | |
| Female | Ref | | Ref | |
| male | 0.88(0.56, 0.39) | 0.585 | 0.60 (0.35, 1.03) | 0.06 |
| **PLWH** | | | | |
| No | ref | | ref | |
| Yes | 1.78 (1.09, 2.92) | **0.021** | 1.75 (1.00, 3.08) | **0.049** |
| **DOT plan** | | | | |
| Observed daily at clinic | Ref | | Ref | |
| Observed daily by family | 0.29(0.17, 0.50) | **<0.001** | 0.24(0.13, 0.45) | **< 0.001** |
| Weight at the start of treatment | 1.00 (0.99 1.01) | 0.83 | 1.00 (0.99, 1.01) | 0.449 |

**Abbreviation:** DOT (Direct observation therapy, PLWH (People living with HIV)

Our study identified a higher mortality rate among adult patients living with HIV compared to those without HIV. This finding aligns with research from other African countries, including studies conducted in Zambia (2021), South Africa (2022), and Kenya (2023), which all reported a similar association between HIV and increased mortality rates [12, 18, 19]. HIV targets CD4+ T cells, and their loss results in an impaired immune response, reducing the effectiveness of both the innate and adaptive immune systems [20]. This, in turn, can lead to new TB infections or the reactivation of latent TB infections, which can result in active TB disease [20]. Consequently, this dual burden leads to a higher mortality rate [21].

This study also found that both pediatrics and adult participants with TB who received treatment through a facility- based direct observation plan (DOT) had a likelihood of dying when compared to those on home-based DOT plan. This finding appears to contradict a Zambian Study (2023) which reported improved survival rates for patients on facility-based DOT

**Table 4. Univariable and multivariable logistic regression of factors associated with drug-susceptible TB mortality among pediatric patients.**

| Variable | OR (95%) | P-value | AOR (95%, Cl) | P-value |
|---|---|---|---|---|
| Age, years | | | | |
| 0–4 | ref | | ref | |
| 5–11 | 1.3 (0.2, 6.8) | 0.79 | 1.13 (0.18, 7.1) | 0.890 |
| 12–18 | 4.5 (1.2, 16.0) | **0.018** | 4.9 (1.2, 19.3) | **0.021** |
| **Sex** | | | | |
| Female | ref | | ref | |
| Male | 0.93 (0.3, 2.9) | | 1.3 (0.4, 4.5) | 0.698 |
| **PLWH** | | | | |
| No | ref | | ref | |
| Yes | 4.6 (1.4, 14.5) | **0.01** | 5.6(1.5, 20.6) | 0.050 |
| **DOT plan** | | | | |
| Observed daily at clinic | Ref | | ref | |
| Observed daily by family | 0.4 (0.08, 1.79) | 0.223 | 0.2 (0.03, 0.99) | **0.039** |

**Abbreviation**: DOT (Direct observation therapy, PLWH (People living with HIV)

compared to those receiving care with home-based observers [22]. Possible explanations for the difference in our findings could be due to resource limitations within our healthcare setting. Shortage of staff and logistical challenges, such as transportation difficulties, might hinder our ability to ensure consistent and effective adherence support within a facility-based DOT program.

This study has several limitations. The study did not collect important data such as HIV-RNA viral load, and ART regimen, nor did it capture relevant social and economic factors like education level, smoking status, alcohol use, employment, and marital status. This limits our ability to fully understand the factors contributing to tuberculosis-related death. Secondly, since the hospital register only records death without cause, some death classified as tuberculosis-related death might have been attributed to other causes due to the absence of autopsy results. However, the study also has a significant strength. The large sample size allows for more generalizable insights into the prevalence of tuberculosis-related death and the factors associated with it.

## Conclusion

This study assessed the prevalence of tuberculosis-related deaths and the associated factors among patients at the Livingstone University Teaching Hospital chest clinic. The findings revealed a high mortality rate which significantly exceeds the national target of less than 5%. Several factors were identified as contributing to the increased mortality among TB patients in our setting which included older age in pediatric patient, living with HIV in adult patients and Facility based direct observation therapy in both cohorts. Addressing TB mortality in this context requires a multifaceted approach, there is need to provide targeted diagnosis and treatment among older patients and people living with HIV as well as enhance support to facility-based DOT programs by increasing health care personnel staffing and providing transportation assistance for follow-up appointments for patients on TB treatment.

## Supporting information

**S1 Checklist. STROBE checklist.**
(DOCX)

**S1 Table. Basic demographic and clinical characteristics among adult patients with drug-susceptible TB mortality with pulmonary TB and disseminated TB.**
(DOCX)

**S2 Table. Basic demographic and clinical characteristics among pediatrics patients with drug-susceptible TB mortality with pulmonary TB and disseminated TB.**
(DOCX)

**S3 Table. Univariable and multivariable logistic regression associated with mortality among adult patients with pulmonary TB and disseminated TB.**
(DOCX)

**S4 Table. Univariable and multivariable logistic regression associated with mortality among pediatric patients with pulmonary TB and disseminated TB.**
(DOCX)

**S1 Data. Minimal dataset.**
(XLSX)

## Acknowledgments

The authors would like to thank Livingstone University Teaching hospital management for having granted permission to conduct the study at Chest clinic and HAND research group for assistance rendered during the study.

## Author Contributions

**Conceptualization:** Lukundo Siame, Eemmanuel Chembe, Benson M. Hamooya, Sepiso K. Masenga.

**Data curation:** Lukundo Siame, Eemmanuel Chembe, Lweendo Muchaili, Benson M. Hamooya, Sepiso K. Masenga.

**Formal analysis:** Lukundo Siame, Eemmanuel Chembe, Benson M. Hamooya, Sepiso K. Masenga.

**Investigation:** Sepiso K. Masenga.

**Methodology:** Sepiso K. Masenga.

**Project administration:** Sepiso K. Masenga.

**Resources:** Sepiso K. Masenga.

**Supervision:** Sepiso K. Masenga.

**Validation:** Sepiso K. Masenga.

**Visualization:** Sepiso K. Masenga.

**Writing – original draft:** Lukundo Siame, Eemmanuel Chembe, Sepiso K. Masenga.

**Writing – review & editing:** Lweendo Muchaili, Benson M. Hamooya, Sepiso K. Masenga.

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
