## [Decision Letter · Decision Letter 0]

25 Jul 2024

PGPH-D-24-01455

Tuberculosis-related deaths at a tertiary hospital in Zambia: Insights into the prevalence and associated factors

Dear Dr. Masenga,

Thank you for submitting your manuscript to PLOS Global Public Health. After careful consideration, we feel that it has merit but does not fully meet PLOS Global Public Health’s publication criteria as it currently stands. Therefore, we invite you to submit a revised version of the manuscript that addresses the points raised during the review process.

We look forward to receiving your revised manuscript.

Kind regards,

Joseph Baruch Baluku, MMed

Academic Editor

Journal Requirements:

1. We note that your Data Availability Statement is currently as follows: The data underlying the results presented in the study have been provided as

supporting information S2.

Additional Editor Comments (if provided):

Dear Authors,

The comments from the second reviewer can be found in the attachment. Address them as well. You can ignore the concerns on novelty and the need to focus on one form of tuberculosis.

Reviewers' comments:

Reviewer's Responses to Questions

**Comments to the Author**

1. Does this manuscript meet PLOS Global Public Health’s publication criteria? Is the manuscript technically sound, and do the data support the conclusions? The manuscript must describe methodologically and ethically rigorous research with conclusions that are appropriately drawn based on the data presented.

Reviewer #1: Yes

2. Has the statistical analysis been performed appropriately and rigorously?

Reviewer #1: Yes

3. Have the authors made all data underlying the findings in their manuscript fully available (please refer to the Data Availability Statement at the start of the manuscript PDF file)?

Reviewer #1: Yes

4. Is the manuscript presented in an intelligible fashion and written in standard English?

Reviewer #1: Yes

5. Review Comments to the Author

Reviewer #1: Thanks for allowing me to review this interesting manuscript. It is a well writted manuscript

I have the following minor comments

1.Line 70. specify that cotrimoxazole was used for only those patients who are HIV positive

2. Cross check line 110 to 111 for correct sentence construction ..... then excel...the cleaned....

3.Include a figure showing data abstraction flow

4. Line 140; a sentence should not start with a figure

5.The reference study cited in line 173 was done in 2021 not 2024

6. PLOS authors have the option to publish the peer review history of their article (what does this mean?). If published, this will include your full peer review and any attached files.

**Do you want your identity to be public for this peer review?** For information about this choice, including consent withdrawal, please see our Privacy Policy.

Reviewer #1: No

---

## [Editor Report · Decision Letter 1]

13 Sep 2024

Tuberculosis-related deaths at a tertiary hospital in Zambia: Insights into the prevalence and associated factors

PGPH-D-24-01455R1

Dear Prof. Masenga,

We are pleased to inform you that your manuscript 'Tuberculosis-related deaths at a tertiary hospital in Zambia: Insights into the prevalence and associated factors' has been provisionally accepted for publication in PLOS Global Public Health.

Best regards,

Joseph Baruch Baluku, MMed

Academic Editor